# Papaverine: A Miraculous Alkaloid from Opium and Its Multimedicinal Application

**DOI:** 10.3390/molecules28073149

**Published:** 2023-03-31

**Authors:** Sania Ashrafi, Safaet Alam, Arifa Sultana, Asef Raj, Nazim Uddin Emon, Fahmida Tasnim Richi, Tasnuva Sharmin, Myunghan Moon, Moon Nyeo Park, Bonglee Kim

**Affiliations:** 1Department of Pharmaceutical Chemistry, University of Dhaka, Dhaka 1000, Bangladesh; 2Drugs and Toxins Research Division, BCSIR Laboratories Rajshahi, Bangladesh Council of Scientific and Industrial Research, Rajshahi 6206, Bangladesh; 3Department of Pharmacy, Faculty of Science and Engineering, International Islamic University Chittagong, Chittagong 4318, Bangladesh; 4Department of Chemistry and Biochemistry, Cell and Molecular Biology Program, University of Arkansas, Fayetteville, AR 72701, USA; 5Department of Pathology, College of Korean Medicine, Kyung Hee University, Seoul 02447, Republic of Korea

**Keywords:** *Papaver somniferum*, opium, benzylisoquinoline, papaverine, alkaloid, antiviral, anticancer, SARS-CoV-2

## Abstract

The pharmacological actions of benzylisoquinoline alkaloids are quite substantial, and have recently attracted much attention. One of the principle benzylisoquinoline alkaloids has been found in the unripe seed capsules of *Papaver somniferum* L. Although it lacks analgesic effects and is unrelated to the compounds in the morphine class, it is a peripheral vasodilator and has a direct effect on vessels. It is reported to inhibit the cyclic adenosine monophosphate (cAMP) and cyclic guanosine monophosphate (cGMP) phosphodiesterase in smooth muscles, and it has been observed to increase intracellular levels of cAMP and cGMP. It induces coronary, cerebral, and pulmonary artery dilatation and helps to lower cerebral vascular resistance and enhance cerebral blood flow. Current pharmacological research has revealed that papaverine demonstrates a variety of biological activities, including activity against erectile dysfunction, postoperative vasospasms, and pulmonary vasoconstriction, as well as antiviral, cardioprotective, anti-inflammatory, anticancer, neuroprotective, and gestational actions. It was recently demonstrated that papaverine has the potential to control SARS-CoV-2 by preventing its cytopathic effect. These experiments were carried out both in vitro and in vivo and require an extensive understanding of the mechanisms of action. With its multiple mechanisms, papaverine can be considered as a natural compound that is used to develop therapeutic drugs. To validate its applications, additional research is required into its precise therapeutic mechanisms as well as its acute and chronic toxicities. Therefore, the goal of this review is to discuss the major studies and reported clinical studies looking into the pharmacological effects of papaverine and the mechanisms of action underneath these effects. Additionally, it is recommended to conduct further research via significant pharmacodynamic and pharmacokinetic studies.

## 1. Introduction

Since ancient times, medicinal plants have played a key role in traditional medicine systems. Phytochemicals are being mined more frequently to find novel leads in the drug discovery process or to find better alternatives to existing ones [1,2,3,4,5]. Around 75% of the global population, mostly from developing countries, depend primarily on traditional herbal medicines due to their affordability and environmentally beneficial qualities [6].

According to estimates, 20% of plant species produce 12,000 alkaloids combined, many of which have been used in both traditional and modern-day medicine for centuries. Among these, about 2500 of the substances are members of a structurally diverse class of metabolites known as benzylisoquinoline alkaloids (BIAs), which also includes the opiate drugs morphine and codeine. Additionally, the benzylisoquinoline alkaloid family is a prominent class of plant-derived chemicals that has shown a wide range of pharmacological activity, including antibacterial, antitussive, antispasmodic, and anticancer properties [7,8].

A prominent benzylisoquinoline alkaloid is papaverine, which can be obtained from *Papaverine somniferum* L. (opium poppy). The opium alkaloids include papaverine, morphine, codeine, thebaine, noscapine, and narceine, as well as a small percentage of some other compounds. Various pieces of traditional research evidence have demonstrated opium alkaloids in Chinese and Indian herbal medicine to be effective at treating a variety of ailments, including chronic cough, rectum prolapse, diarrhea, dysentery, and gastrointestinal issues. In addition, papaverine has also been incorporated in therapeutic settings to treat erectile dysfunction, smooth muscle spasms, and spasms associated with gastrointestinal problems. Scientists have also found papaverine as a nonselective phosphodiesterase (PDE) inhibitor in mammals, boosting the amount of cAMP and cGMP available for cell signaling [9]. Hence, this secondary metabolite demands further exploration and requires pharmacological research investigations. Therefore, the current study focuses on the molecular mechanisms of papaverine’s pharmacological potential as they have been investigated in diverse experimental models.

## 2. Natural Source of Papaverine

Due to their phytochemical composition, members of the genus *Papaver* (family: Papaveraceae) are recognized for their therapeutic benefits. The most significant *Papaver* species that contributes phytochemicals for drug development is *Papaver somniferum* L. (opium poppy), which is highly produced in countries such as Afghanistan, Myanmar, Mexico, Laos, Turkey, Czechia, and Spain. Other commonly cultivated *Papaver* species include *P. bracteatum* Lindl. (Persian poppy), *P. rhoeas* L. (common poppy or corn poppy), *P. dubium* L., *P. pseudo-orientale* Medw., and *P. orientale* L., which are grown at high altitudes in north and northwest Iran, Russia, the Caucasia region, Europe, and America [10]. *P. somniferum* L. produces papaverine naturally in its unripe seed capsules. A total of 40 alkaloids have been found in the plant; however, morphine (10–15%), noscapine (4–5%), codeine (1–3%), papaverine (1–3%), and thebaine (1–3%) are the five primary alkaloids. The prevalence of papaverine in Indian species ranges from 0.5% to 3% [11].

## 3. Chemistry of Papaverine

Benzylisoquinoline alkaloids hold a prominent place in alkaloid chemistry as they serve as in vivo precursors to many other naturally occurring isoquinolines. They are either 1,2,3,4-tetrahydro, as in coclaurine and *N*-nororientaline, or fully aromatic, as in papaverine, palaudine, and escholamine. Ring A in the benzylisoquinoline alkaloids may possess two or three oxygenated substituents, while ring C has only one or two substituents [12].

Papaverine, also known by its IUPAC nomenclature 1-[(3,4-dimethoxy phenyl) methyl]-6,7-dimethoxyisoquinoline, is one of the principal benzylisoquinoline alkaloids found in *P. somniferum* [13,14]. Naturally, it is produced as a byproduct of morphine, codeine, and narcotine synthesis. Its *m*/*z* ratio was determined to be 340.15417 [15]. It is a neutral solid that only slightly dissolves in water [16]. There are four methoxy groups in papaverine. Even if the molecule lacks a TV-methyl group, it still functions as a tertiary base. It is considered a pyridine derivative because it may be reduced to a secondary amine by adding four hydrogen atoms, with the heterocyclic ring fused to a benzene ring [12] (Figure 1).

Guido Goldschmiedt first illustrated the structure of the papaverine between the years of 1885 and 1898. By forming methiodide and demonstrating the presence of four methoxy groups per mole, he established the existence of a tertiary nitrogen atom. Under different conditions, the base was oxidized with potassium permanganate to produce various related compounds (Figure 2).

The current structure was recognized as papaverine based on the evidence mentioned above and other relevant data. Following the successful synthesis of papaverine in 1909, Pictet and Gams validated the molecular structure [14].

## 4. Biosynthesis of Papaverine

Two units of tyrosine contribute as the precursors for the biosynthesis of papaverine, and the intermediate products include (*S*)- norcoclaurine, laudanine, norlaudanine, reticuline, norreticuline, tetrahydropapaverine, and dihydropapaverine. Recent investigations revealed that the primary pathway of papaverine biosynthesis in the opium poppy has been identified by systematic silencing of benzylisoquinoline alkaloid biosynthetic genes.

There are two suggested metabolic pathways for (*S*)-norcoclaurine. One involves only *N*-demethylated intermediates (the NH pathway), whereas the other involves (*S*)-reticuline and involves a number of *N*-methylated intermediates (the NCH_3_ pathway) [13,17,18]. The NH route advances via (*S*)- norreticuline [13,19,20] (Figure 3), whereas the NCH_3_ route involves (*S*)-reticuline [13,15,20,21] (Figure 4).

The first step in papaverine biosynthesis is the condensation of two L-tyrosine derivatives, 4-hydroxyphenylacetaldehyde (4HPAA) and dopamine, which is accomplished through decarboxylation, meta-hydroxylation, and transamination to produce the precursor to all other benzylisoquinoline alkaloids, (*S*)-norcoclaurine. Tyrosine decarboxylase (TYDC) and tyramine 3-hydroxylase (3OHase) transform L-tyrosine into tyramine and dopamine, respectively. L-tyrosine can be transaminated by L-tyrosine aminotransferase (TyrAT) in the production of 4HPAA, and then decarboxylated by an enzyme identified as 4-hydroxyphenylpyruvate decarboxylase (4HPPDC). Norcoclaurine synthase (NCS) is the enzyme that catalyzes the condensation of (*S*)-norcoclaurine from 4HPAA and dopamine.

Norcoclaurine-6-*O*-methyltransferase (6OMT) first transforms (*S*)-norcoclaurine into (*S*)-coclaurine. In the NH pathway, (*S*)-coclaurine first undergoes 3′ hydroxylation by 3′ hydroxylase (3′OHase) and then is converted to (*S*)-norreticuline by 3′-*O*-methyltransferase (3′OMT). On the other hand, in the NCH_3_ pathway, (*S*)-coclaurine is taken up by coclaurine *N*-methyltransferase (CNMT) to yield (*S*)-*N*-methylcoclaurine. (*S*)-*N*-methylcoclaurine is hydroxylated to 3′-hydroxy-*N*-methylcoclaurine by (*S*)-*N*-methylcoclaurine 3′-hydroxylase (NMCH), which is then transformed into (S)-reticuline by 3′-hydroxy-*N*-Methylcoclaurine 4′-*O*-methyltransferase (4′OMT). It is interesting to note that only NMCH has been reported to exhibit strict stereoisomer and substrate specificity, accepting only (*S*)-*N*-methylcoclaurine and rejecting either the corresponding (*R*)-*N*-methylcoclaurine or N-desmethyl compounds. As a distinction, the O- and *N*-methyltransferases often accept a wide range of (*R*)- and (*S*)-tetrahydroisoquinolines [19]. The enzyme reticuline 7-*O*-methyltransferase (7OMT) can further methylate reticuline to produce laudanine, which can then be fully *O*-methylated to laudanosine by 3′-*O*-methyltransferase.

The final steps in papaverine biosynthesis comprise the oxidation of the fully *O*-methylated and N-desmethyl molecule tetrahydropapaverine by dihydrobenzophenanthridine oxidase (DBOX).

Advanced quantum chemical density functional theory (DFT) calculations, as well as diffuse reflectance (Ds), experimental electronic absorption (EAs), matrix-associated laser desorption ionization (MALDI) coupled with Orbitrap imaging mass spectrometry (MS), fluorescence spectroscopy (Fs), and circular dichroic (CD) have been used for theoretical and experimental elucidation of the papaverine biosynthetic pathway via (S)-reticuline (the NCH_3_ pathway) [19,20,22].

## 5. Mechanism of Action of Papaverine

Papaverine is recognized as the most effective smooth muscle relaxant, as it acts directly on smooth muscle by exerting a strong vasodilating effect. It has been observed to boost intracellular levels of cAMP and cGMP by blocking the cAMP and cGMP phosphodiesterase in smooth muscles (Figure 5) [23,24,25,26,27]. Inhibiting the release of calcium from the intracellular space and obstructing calcium ion channels in the cell membrane are two other ways that papaverine may work [28].

## 6. Pharmacological Properties of Papaverine

Papaverine has been demonstrated to be a particular PDE_10_A phosphodiesterase inhibitor, which is mostly found in the striatum of the brain. The chronic injection of it into mice resulted in motor and cognitive deficits as well as elevated anxiety. Other studies have suggested that it may also have an antipsychotic effect. However, not all research has supported this theory [29,30]. Nevertheless, papaverine has been approved for the treatment of GIT, bile duct, and ureter spasmolytic disorders [31].

### 6.1. Activity against Erectile Dysfunction (ED)

PDE_5_ inhibitors are used as the first-line therapy for ED [32]. As a popular vasodilator, their usage has been observed to improve penile impotence [33]. Its ability to treat ED and impotence has been known for a long time. By far, the largest number of studies have been published demonstrating the effectiveness of papaverine in treating erectile dysfunction (ED) and impotence [34]. Three simultaneous and synergistic processes work together to maintain normal erectile function: (1) relaxation of the cavernosal smooth muscle, (2) an increase in penile arterial inflow caused by neurological activity, and (3) a restriction of venous outflow from the penis. These processes occur due to the following: (1) the activation of cGMP-dependent protein kinase G (PKG); (2) the activation of cGMP-dependent ion channels that reduce intracellular Ca^2+^ by Ca^2+^ sequestration and/or extrusion; (3) the opening of K^+^ channels, causing the hyperpolarization of corpus cavernosum smooth muscle cells; and (4) the activation of myosin light-chain phosphatases. The objective of ED pharmacotherapy is to develop novel pharmacological targets that inhibit the contractile systems (α-adrenoceptor antagonists) and activate (e.g., prostaglandin E_1_ (PGE_1_), NO-donors, and forskolin) or augment (e.g., PDE inhibitors and gene therapy) the vasodilatory systems to produce greater trabecular smooth muscle relaxation of the corpora cavernosa [35,36,37] (Figure 6).

Intracavernous papaverine was found to play a vital role in the management of male erectile failure in a study on 48 patients with psychogenic impotence. Intracavernous papaverine induces erection by several mechanisms. It relaxes the smooth muscles of sinusoids and increases the arterial flow to the corpora. The use of papaverine has also been linked to increased venous outflow resistance [34,38]. Research conducted on 17 men with organic impotence revealed papaverine gel to cause a noticeably larger cavernous artery diameter [39]. Although its use as a monotherapy to treat impotence was initially questioned due to priapism being a significant side effect (which occurs in 15–18% of patients), physicians soon discovered that this side effect was dose-dependent and only occurs in patients with neurogenic impotence [40,41]. To lessen toxicity and priapism, papaverine was combined with phentolamine and PGE_1_ [35]. By raising the amount of intracellular cyclic adenosine monophosphate, relaxing the smooth muscle of the cavernous body and helicine arteries, and inhibiting the enzyme phosphodiesterase, papaverine and phentolamine were able to significantly increase erections in various experiments [42,43]. It has been observed that papaverine’s effectiveness is equal to that of oral sildenafil in a different trial that involved 31 male patients who had ED injuries and were in the early stages of paraplegia [44]. Recently, it was discovered that a new topical formulation using lyotropic liquid crystal (LLC) systems and papaverine-HCl was a suitable and efficient substitute for the injectable formulation in the treatment of ED [34]. According to a study conducted in vitro, the substance increases the motility of post-thaw sperm [34].

### 6.2. Activity against Pulmonary Vasoconstriction

The protective effects of papaverine on the lungs have been demonstrated through various mechanisms. In a model of pulmonary embolism caused by autologous blood clots in rabbit lungs that were isolated and perfused, papaverine was discovered to lessen pulmonary vasoconstriction and edema. In the pulmonary vascular bed, papaverine can diminish the vasoconstrictor response to ET-1, TxA_2_, and serotonin without altering their release. It is generally known that PDE inhibitors also have antiplatelet effects due to their ability to raise cAMP levels. Papaverine, in particular, has been demonstrated to reduce platelet aggregation brought on by ADP as well. Such an outcome might have contributed to the better results observed in the papaverine-treated group in [45]. It was discovered that it inhibits voltage-gated Ca^2+^ channels in a concentration-dependent manner, resulting in the relaxation of tracheal smooth muscle [46]. Papaverine with nifedipine effectively decreased the pulmonary artery vasoconstriction brought on by ECS (Euro-Collins solution). The elevation in cAMP caused by papaverine may improve lung preservation. Additionally, it has been noted to reduce Ca^2+^ influx via cell membranes [34].

### 6.3. Postoperative Vasospasm

To significantly minimize postoperative vasospasms and maintain regular vascular morphology throughout antispasmodic therapy, papaverine-loaded electrospun fibrous membranes were developed [47]. In a rabbit model, it was discovered that the intra-arterial route is more effective for lowering autologous blood-induced cerebral vasospasms [48]. For a very long time, papaverine has been used to prevent vasospasms induced by subarachnoid hemorrhage [49]. In numerous investigations, including in patients who had aneurysmal subarachnoid hemorrhages, the effectiveness of papaverine in avoiding vasospasms was validated. Papaverine can be given either on its own or in conjunction with transluminal balloon angioplasty. In these circumstances, papaverine has been found to improve cerebral oxygenation, increase the angiographic vessel diameter, decrease the extended cerebral circulation time, and boost cerebral blood flow in an effort to prevent cerebral infarction. Here, a significant barrier is the short-lived nature of papaverine [50,51,52,53]. A sustained-release formulation that can be implanted intracranially might minimize this, and it would also lower the likelihood of hypotension during surgery [54]. Nevertheless, a different trial involving 31 patients with a subarachnoid hemorrhage-related vasospasm found no additional benefits from papaverine when compared to the medical treatment of vasospasms alone. The authors came to the conclusion that changing the time or indications for therapeutic intervention could be advantageous [49]. Another retrospective study on nine consecutive patients with acute large-artery occlusion treated with a stent retriever and intra-arterial papaverine demonstrated an increase in the caliber and flow of the infused arteries, suggesting a safe and effective method of treating cerebral vasospasms following mechanical thrombectomy in acute ischemic stroke [55]. Another investigation of 27 patients with a subarachnoid hemorrhage-related symptomatic vasospasm discovered that intra-arterial papaverine consistently reduces cerebral circulation time [56]. Following intra-arterial infusion of papaverine, individuals with symptomatic vasospasms showed an improvement in cerebral oxygenation as well as a reduction in cerebral lactic acidosis [57].

### 6.4. Antiviral Properties

Papaverine is also recognized for its antiviral activities against different human viruses and the murine retrovirus, MSV-Harvey. It has been hypothesized that, at least for the measles virus, interference with cellular DNA synthesis directly, competitive and reversible binding to the DNA molecule, or an increase in endogenous cAMP will impede viral RNA synthesis and the phosphorylation of viral proteins [58]. In a study, papaverine suppressed the viral growth of measles in neuroblastoma cells by inhibiting the synthesis of viral RNAs in a dose-dependent and reversible manner [59]. It was reported to show effective antiviral activity by inhibiting the replication of the CMV virus. The mechanism of action underlying the relaxing effect of these drugs on smooth muscle may prevent at least the initial cell rounding, and it is possible that a critical physiologic event(s) (e.g., the rise in intracellular free Ca^2+^) may be important to both early cellular responses and CMV replication [60]. Papaverine inhibited the replication of HIV in H9 cell lines by blocking the RT activity and p24 expression. It also showed inhibiting activity in the peripheral blood mononuclear cell (PBMC) culture by influencing the viral markers RT and p24 [58,61]. It inhibited the replication of the measles virus in neural cells [62]. Papaverine showed a dose-dependent inhibition of multiple strains of influenza virus when A/WSN/33 (H_1_N_1_), A/Udorn/72 (H_3_N_2_), and B/Lee/40 were used in a study [63]. In a very recent study, papaverine revealed its ability to inhibit the SARS-CoV-2 cytopathicity in the human epithelial colorectal adenocarcinoma cell line, Caco-2 [64] (Table 1).

### 6.5. Cardiovascular Activity

Papaverine exhibited potent cardioprotective effects by diverse mechanisms. It directly stimulated the sinus rate and atrial contractility by demonstrating positive chronotropic and inotropic effects on an isolated atrial preparation from a dog, which again points to the inhibition of PDE and accumulated cAMP. Additionally, papaverine may partially trigger the release of catecholamines from adrenergic nerve fibers and may interfere with the process of adenosine uptake. It is hypothesized that papaverine may directly stimulate atrial contractility and SA nodal pacemaker activity [65]. Similar effects were found in another study where papaverine displayed positive inotropic effects on atrial preparation, whereas in ventricular preparation, it did not affect the force of contraction significantly [66]. Papaverine inhibits both hKv1.5 and native hKv1.5 channels in a concentration, voltage, state, and time-dependent manner. This interaction shows that papaverine may change cardiac excitability in vivo [67].

### 6.6. Anti-Inflammatory Activity

Through the cAMP/PKA and MEK/Erk pathways, papaverine reduced the expression of proinflammatory factors and inhibited the activation of primary retinal microglia caused by LPS, and the MEK/Erk pathway may be partially regulated by cAMP/PKA, which can provide theoretical and experimental support for its protection of the central nervous system [68]. Yoshikawa et al. first noticed that papaverine could prevent the release of TNF-α and IL-1β in LPS-induced BV2 cells [69]. Similar activities were reported in another study where papaverine prevented the production of nitric oxide and proinflammatory cytokines in LPS-stimulated microglia [31]. Furthermore, it appeared to have anti-inflammatory effects in mouse models by inhibiting high mobility group box 1-mediated inflammatory responses [70]. Similar effects were demonstrated in LPS-stimulated macrophages and microglia where papaverine suppressed TNF-α [68,71], IL1β, and the NF-κB signaling pathway [72], thus proving its potential to treat neurodegenerative diseases.

### 6.7. Anticancer Activity

It was found that papaverine effectively induced a morphological change and inhibited proliferation and the invasive potential of human prostate cell lines PC-3, DU145, and LNCaP primarily through its PDE-inhibiting capability, which resulted in raised cAMP levels [73]. Similar effects were reported on the LNCaP cell line due to a synergistic effect induced by a combination of papaverine and prostaglandin E_2_ (PGE_2_) [74] and on PC-3 by inducing apoptosis and cell cycle arrest along with the downregulation of NFkB and the PI3K/Akt signaling pathway [75]. The phytochemical has been reported to exhibit cytotoxic effects on cancerous HT29, T47D, and HT1080 cell lines without affecting the noncancerous mouse NIH_3_T_3_ cell line as compared to doxorubicin, a widely used anticancer drug. The mechanism behind it was selective DNA damage and the induction of apoptosis on cancerous cell lines [76]. It expressed a cytotoxic effect against cancer stem cells, especially human breast cancer cell line MCF-7, by arresting the cell cycle in the G_1_ phase and inducing apoptosis [77]. Antiproliferative activity of the compound was reported on hepatocarcinoma cell line HepG-2 as it affected the telomerase activity [78]. It has been proven to be an effective radiosensitizing agent that reduces the rate of oxygen consumption through the inhibition of mitochondrial complex I. Thus, the compound was found to improve the response to radiation therapy and is a potential candidate for tumor hypoxia treatment [9]. Papaverine is found to prevent cell migration and delay zebrafish development by suppressing the kit-signaling pathway [79]. The compound significantly inhibited the proliferation of human glioblastoma cell lines U87MG and T98G and the tumor volume in the U87MG xenograft mouse model [80,81]. The papaverine–Au(III) complex was reported to have better cytotoxic activities against human breast cancer MCF-7 cells and hepatocellular carcinoma HepG-2 cells than papaverine itself, and the inhibiting ability was higher than that of cisplatin against MCF-7 [16]. Caroverine, which is one derivative of papaverine, prevented the expression of VEGF, which is a well-known tumor-promoting factor [82]. In another study, a papaverine oxidation product that is a 6a,12a-diazadibenzo-[a,g]fluorenylium derivative inhibited the MCF-7 cell line by blocking the G0/G1 phase of the cell cycle and telomerase activity [83]. An investigation involving *S. cerevisiae* and docking and molecular dynamic simulation studies showed evidence that papaverine induces ROS-mediated apoptosis and inhibits Bcr-Abl downstream signaling [84] (Table 2).

### 6.8. Neuroprotective Effect

Papaverine may also exert neuroprotective effects to treat neuropsychiatric diseases such as schizophrenia and depression [85,86]. In a clinical trial involving three female patients with tardive dyskinesia, it was revealed that daily 300 to 600 mg doses of papaverine improved the dyskinesia condition without showing any side effects. The authors concluded that the effects were due to the inhibition of the dopamine pathway, which might be the reason for dyskinetic movements [87]. The study was again conducted with a larger number of patients via the oral administration of sustained-release 150 mg papaverine capsules. Two out of nine patients showed clinical improvements [88]. Oro-facial dyskinesia was improved in another clinical trial conducted on 150 patients [89]. Papaverine was also found to potentiate nerve growth factor (NGF)-induced neurite outgrowth in PC12 cells in a concentration-dependent manner [90]. By significantly raising the levels of BDNF, synapsin-IIa, DCX, pCREB, IL-10, and GSH in various brain regions while significantly lowering the levels of TNF-α, IL-6, and TBARS, the drug was found to restore the basic behavioral phenotype in autism spectrum disorder [9]. Another study revealed that the substance may be useful in reducing the ischemic infarct volume, suggesting that it may be used to treat cerebral ischemia in clinical practice [91]. By modifying the NF-B and CREB signaling pathways, it prevents the activation of the NLRP_3_ inflammasome, which reduces microglial activation and neuronal cell death. As a result, it could be a promising treatment for Parkinson’s disease, which is exacerbated by systemic inflammation [92]. In the subacute MPTP/P animal model of Parkinson’s disease, the data revealed that papaverine reduces neuroinflammation and MMP-3 production, which prevents dopaminergic neuronal cell death and α-synuclein aggregation. In light of this, it might be a viable medication for the management of Parkinson’s disease [93]. Papaverine enhanced cognitive function in a mouse model with Huntington’s disease by inhibiting PDE_10_, resulting in cAMP-responsive element-binding protein (CREB) phosphorylation and GluA1 [94]. It also provided efficient protection to the spinal cord during descending thoracic and thoracoabdominal aortic aneurysm repair surgery by perfusing the spinal cord [1]. In another study, papaverine revealed its direct effect on synaptic vesicles, which was exhibited via the increase in norepinephrine and dopamine-β-hydroxylase from isolated perfused cat spleen [95]. Furthermore, papaverine temporarily increased sublingual microcirculatory blood flow in septic shock patients who needed vasoconstrictors to maintain blood pressure during fluid resuscitation without affecting systemic hemodynamics [96].

### 6.9. Gestational Activity

In the 1990s, research on papaverine and its derivatives revealed that they could shorten the time needed for the first stage of labor. A specific inhibitor of PDE 4, drotaverine hydrochloride, is a homolog of papaverine. Because of its ability to relax smooth muscles, it was found to be beneficial in accelerating cervical dilation [97]. Madhu et al. found that women who were treated with drotaverine through the latent phase of labor had a significantly shorter time between the administration of the medication and the delivery of the fetus compared to women who were treated with a placebo. The study involved 146 women who gave birth vaginally (182 min compared to 245 min with a placebo) [98]. Another retrospective comparative investigation of 498 pregnant women indicated that short-term prenatal exposure to papaverine adjusted for indication was not linked to preterm births, cesarean birth, reduced birth weight, small gestational age, or perinatal death [99]. Additionally, it was claimed that the medication worked well to lower pre-eclamptic patients’ blood pressure [100].

### 6.10. Other Activities

Papaverine hydrochloride is equally as effective as sodium diclofenac for the short-term relief of acute renal colic pain, and it may be advantageous in patients with contraindications according to a prospective, single-blind clinical study that involved 86 patients with acute renal colic who were given 120 mg intravenous papaverine hydrochloride [101]. Another study found that the injection of alprostadil and papaverine into the spermatic cord protected against ischemia/reperfusion injury following right-side testes torsion and reduced histological alterations following testicular ischemia-reperfusion injury [102].

## 7. Limitations of the Study

Executing an ideal drug discovery and development process is one of the primary challenges for the pharmaceutical research community. ADME is a critical step in the drug design process that investigates the fate of a drug molecule after ingestion. Notably, drug metabolism studies are critical processes for optimizing the lead compounds with optimal PK/PD features, identifying new chemical entities based on the discovery of active metabolites, minimizing potential safety liabilities due to the development of reactive or toxic metabolites, comparing preclinical metabolism in animals with humans to guarantee that animals used in experiments have the potential to adequately cover human metabolites and support human dose predictions, and so on [103]. However, ADME and the PK/PD parameters of papaverine were not evident in this review. When it comes to drug attrition during the clinical stage of development, compound failure rates because of the toxicity prior to human testing are relatively high, and they may account for up to 30% of the loss. In order to establish an anticipated safe dose range and to gather knowledge on drug distribution, organ-specific toxicity, and metabolism, toxicology studies in at least two nonhuman species are typically utilized [104]. The toxicological parameters of papaverine were not defined in this review. Proper translation and determination of the maximum recommended starting dose in humans is a critical task in new drug development and research [105]. No specific dose of papaverine was studied in this review. Moreover, study data are not available for use in lactating mothers and pediatric and geriatric patients. The three-dimensionality of molecules is intimately related to the clinical success of drug candidates [106], which was not elaborated on in this review. The efficacy of traditional medicines is frequently the consequence of a synergistic interaction between numerous components, targets, and pathways [107]. This review did not include the positive or negative synergistic effects of possible analogs of papaverine found in opium. Possible side effects of papaverine include priapism, penile fibrosis, and arrythmia [108]. No studies have been conducted on whether these side effects can be utilized as a secondary usage via repurposing; e.g., metformin is the first-line therapy of type II diabetes, and it can be repurposed as an antiobesity drug for both diabetic and nondiabetic patients [109,110].

## 8. Discussion and Future Recommendations

Papaverine has been proven to be a high-value opioid alkaloid in the field of therapeutics either in solitude or in combination with other metabolites/molecules [9]. It was approved by the Food and Drug Administration (FDA) of the United States as a vasodilator to be predominantly used in the treatment of cerebral vasospasms and coronary circulation [108]. Several preclinical and clinical studies also demonstrated its potential efficacy against pulmonary vasoconstriction, erectile dysfunction, postoperative vasospasms, some particular viral infections, inflammation, cardiac excitability, carcinoma, neurological disturbances, gestational difficulties, pre-eclampsia, acute renal colic pain, and ischemia-reperfusion injury, as well as other muscle spasm-oriented complications [87,88,91,101]. Some of the notable mechanisms underlying the different pharmacological actions include vasodilation, the activation of cGMP and cAMP-dependent biomolecules, the inhibition of vasoconstrictor responses to biomolecules, interference with certain viral nucleic acids, the inhibition of cytokine release (such as TNF-α, IL-1β, and NF-κB), the apoptosis of diseased cells, the potentiation of neurite outgrowth, the alteration of different biomolecular signaling pathways, etc., which are discussed throughout this review. As a consequence, the multiple bioactive capabilities of papaverine suggest that it may also be an effective natural phytoconstituent in disease management. Moreover, synthetic drugs consist of several drawbacks, such as a lack of bioavailability, cost-effective issues, drug resistance issues, unexpected adverse effects, etc. [111]. To combat these drawbacks, there is a need to search for lead compounds among the natural substances [112]. Plants are a present from the Earth that have been providing us vital phytochemicals for thousands of years [113,114]. Bioactive phytochemicals from natural sources play pivotal roles in drug discovery and development. Almost 80% of all currently available drugs are either directly derived from plant or are a modified version [115,116]. Alkaloids are a very important class of bioactive phytochemicals that play a significant role in drug discovery [117]. Thus, papaverine is a potential natural drug candidate that may be utilized in the near future. Researchers should carry out several studies on the papaverine alkaloid, including by studying the determination and revision of its PK/PD parameters, the therapeutic index, safety and toxicological profiles, dosage, drug–drug interactions, drug-food interactions, and other important parameters. Considering all these factors, papaverine should be subjected to extensive research to establish it as a novel drug and/or lead compound. This review will provide future researchers with important insights for further studies on this conspicuous alkaloid.

## 9. Conclusions

The majority of the alkaloids isolated from the opium poppy seed, such as morphine and codeine, have analgesic properties; nevertheless, papaverine varies from the opium group of alkaloids both chemically and therapeutically. While the majority of the primary alkaloid chemicals derived from the opium poppy are narcotic and have an analgesic effect, the majority of papaverine’s pharmacological usage is as a non-narcotic, non-analgesic smooth muscle relaxant and vasodilator. Papaverine is a recognized inhibitor of phosphodiesterases. Papaverine is FDA-approved and is already in clinical use as a vasodilator. It is gaining more and more attention for use in other biological activities. Within the scope of this work, we described multiple bioactive properties of papaverine in addition to the molecular mechanisms behind such activities. Both in vitro and in vivo as well as clinical studies showed that papaverine possessed considerable pharmacological properties besides its vasodilator effects. As a result, it is a vital potential candidate both for the discovery of novel drugs and the development of the existing drug. As a matter of fact, its antiviral and anticancer actions both exhibit unique mechanisms of action that show considerable potential for treating their respective illnesses, which demonstrates that papaverine is a prominent candidate for use in the research and development of new antiviral and anticancer medications. In addition, toxicological research must be carried out to establish the substance’s safety for use in other pharmaceutical applications.

## Figures and Tables

**Figure 1 molecules-28-03149-f001:**
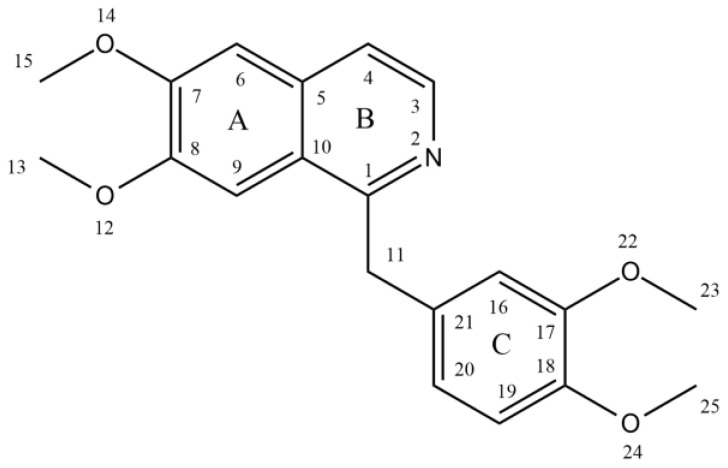
Structure of papaverine.

**Figure 2 molecules-28-03149-f002:**
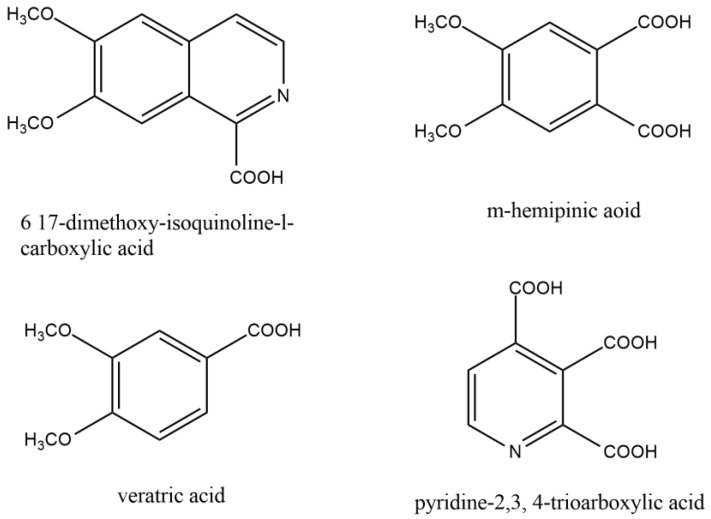
Similar compounds of papaverine helped to elucidate the structure of papaverine.

**Figure 3 molecules-28-03149-f003:**
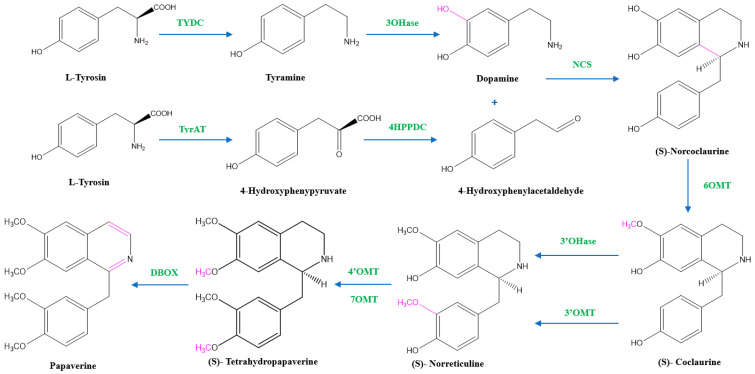
The NH pathway of papaverine synthesis. TYDC = tyrosine decarboxylase, TyrAT = L-tyrosine aminotransferase, 4HPPDC = 4-hydroxyphenylpyruvate decarboxylase, 3OHase = tyramine 3-hydroxylase, NCS = norcoclaurine synthase, 6OMT = norcoclaurine-6-*O*-methyltransferase, 3’OHase = 3’ hydroxylase, 3’OMT = 3′-*O*-methyltransferase, 4’OMT = 3′-hydroxy-*N*-methylcoclaurine 4′-*O*-methyltransferase, 7OMT = norreticuline 7-*O*-methyltransferase, DBOX = dihydrobenzophenanthridine oxidase.

**Figure 4 molecules-28-03149-f004:**
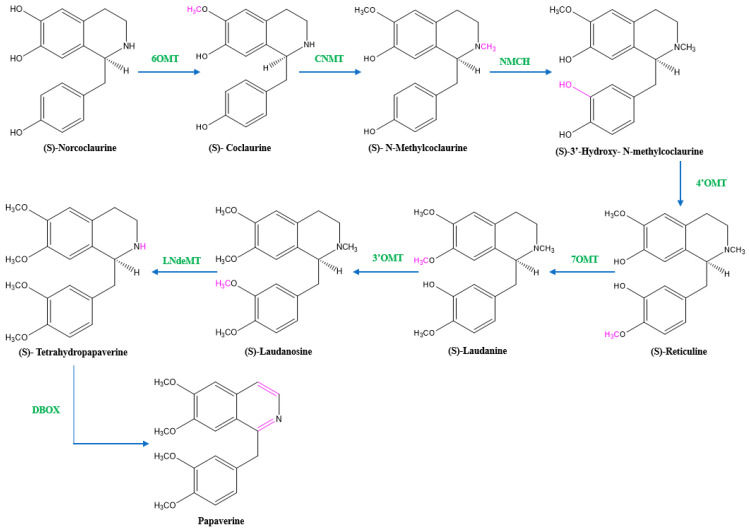
The NCH_3_ pathway of papaverine synthesis. 6OMT = norcoclaurine-6-*O*-methyltransferase, CNMT = coclaurine-*N*-methyltransferase, NMCH = (*S*)-*N*-methylcoclaurine 3′-hydroxylase, 4’OMT = 4′-*O*-methyltransferase, 7OMT = reticuline 7-*O*-methyltransferase, 3’OMT = 3′-*O*-methyltransferase, LNdeMT = laudanosine *N*-demethylase, DBOX = dihydrobenzophenanthridine oxidase.

**Figure 5 molecules-28-03149-f005:**
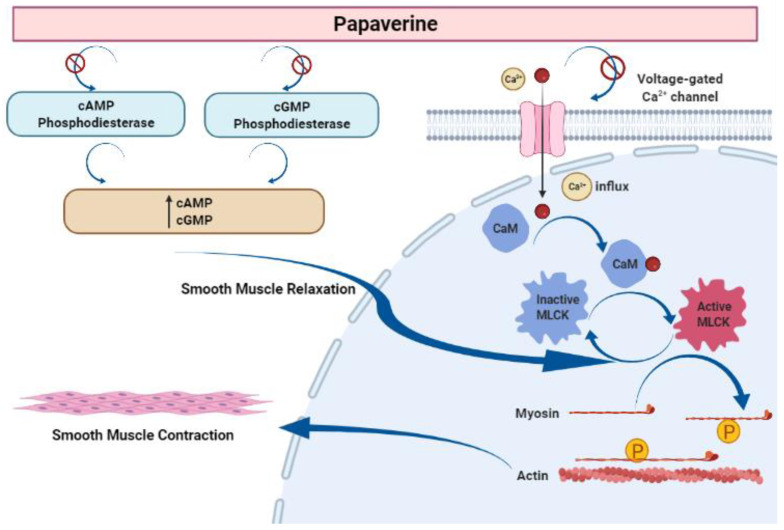
Mechanism of action of papaverine in smooth muscle relaxation. Smooth muscle contraction requires five steps: After the increase in intracellular Ca^2+^ concentration from the extracellular fluid, these ions bind to a protein called calmodulin (CaM). This complex activates a protein called myosin light-chain kinase (MLCK) (papaverine inhibits this step), which subsequently phosphorylates light chains of myosin heads, increasing the myosin ATPase activity. Finally, active myosin cross-bridges slide along actin and create muscle tension to contract the cell.

**Figure 6 molecules-28-03149-f006:**
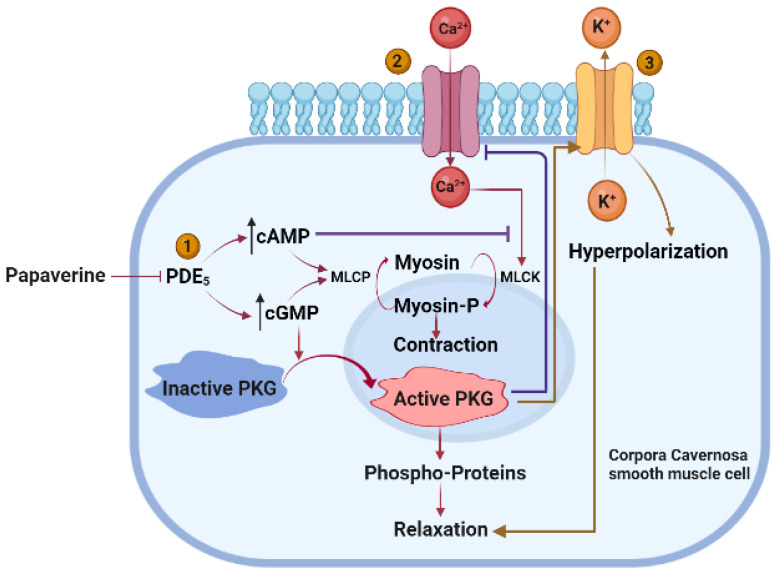
Mechanism of action of papaverine in relaxation of cavernosal smooth muscle. Papaverine blocks cAMP and cGMP phosphodiesterase to raise the concentration of cAMP and cGMP, which further releases MLCP that dephosphorylates myosin, resulting in smooth muscle relaxation and increased cGMP that activates PKG and leads to smooth muscle relaxation (1). Activated PKG lowers Ca^2+^ influx. Ca^2+^ activates MLCK, which contracts smooth muscle via myosin phosphorylation (2). Papaverine induces efflux of K+ with subsequent hyperpolarization and relaxation of corpora cavernosa smooth muscle cells.

**Table 1 molecules-28-03149-t001:** Antiviral properties of papaverine.

Molecule	Activity Against	Experimental Approaches	Key Result	Mechanism of Action	Reference
**Papaverine hydrochloride**	HIV	Determination of viral replication by liquid competition radioimmunoassay in H9 cell line and in peripheral blood mononuclear cell (PBMC) culture.	- The drug at a concentration of 10 μg/mL resulted in no reverse transcriptase activity or p24 expression-specific viral markers in the supernatant and no virus antigen detection at the cellular level. - The drug affected the synthesis of the env precursor protein gpl60. - A marked decrease in the expression of the viral proteins was also observed after treatment with papaverine.	- Interfere with DNA synthesis through competitive and reversible binding to the DNA molecule. - From the data, the authors concluded that papaverine seems to affect the late steps of HIV replication. In fact, the selective effects on different proteins suggest that papaverine acts after reverse transcription.	[58]
		Determination of viral replication in MT4 cell line and in peripheral blood mononuclear cell (PBMC) culture. Examination of T-cell lymphocytes.	Papaverine significantly inhibited HIV replication by more than 99% at doses of 30 μM with an CD_50_ and ED_50_ of 32 μM and 5.8 μM, respectively.	The drug might affect cellular DNA synthesis and reverse transcription, indirectly inhibiting HIV replication.	[61]
**Papaverine**	Measles virus	Determination of viral replication in neural and non-neural cells. Analysis of mechanism for the inhibition of viral replication.	Suppression of virus growth was most prominent in neuroblastoma cells, followed by that in epidermoid carcinoma and glioblastoma cells.	- Synthesis of viral RNAs, including genomic RNA and mRNA, was inhibited. - Phosphorylation of the viral proteins was inhibited.	[59]
**Papaverine**	CMV	Assays for inhibition of infectious CMV yields on human embryo skin-muscle (SM) cells. Assays for the rate of cell DNA synthesis by measuring the incorporation of [methyl^3^H] thymidine into cell DNA.	Inhibition of the multiplication of CMV. Papaverine was the most potent of the three drugs (papaverine, verapamil and sodium nitroprusside); at a concentration of 30 μg/m (80 μM) the CMV yield was inhibited by 5.21 log_10_ at 120 hr postinfection (PI).	- Relaxing effect of papaverine on smooth muscle may at least prevent the initial cell rounding. - The greater potency of papaverine relative to nitroprusside may have resulted from increased levels of both cyclic AMP (cAMP) and cyclic GMP (cGMP) rather than from cGMP alone. - It is possible that a critical physiologic event(s) (e.g., the rise in intracellular free Ca^2+^) may be important to both early cellular responses and CMV replication.	[60]
**Papaverine**	Various strains of influenza virus as well as the paramyxoviruses parainfluenza virus 5 (PIV5), human parainfluenza virus 3 (HPIV3), and respiratory syncytial virus (RSV)	Determination of antiviral activity by plaque reduction neutralization test (PRNT).	Dose-dependent inhibition of influenza virus strains.	- Kinetic studies demonstrated that papaverine inhibited influenza virus infection at a late stage in the virus life cycle through the suppression of nuclear export of vRNP, and also interfered with the host cellular cAMP and MEK/ERK cascade pathways.	[63]
**Papaverine**	SARS-CoV-2	Cytopathicity assays.	Inhibit SARS-CoV-2 cytopathicity in the human epithelial colorectal adenocarcinoma cell line, Caco-2, with IC_50_ value of 1.1 ± 0.39.	Additional studies required.	[64]

**Table 2 molecules-28-03149-t002:** Anticancer properties of papaverine.

Molecule	Cell Line	Cell Type	Significant Benefit Achieved	Reference
**Papaverine**	PC-3, DU145, and LNCaP	Prostate cancer	Induced morphologic change and also raised intracellular cyclic AMP levels in LNCaP cells.	[73]
**Papaverine combined with prostaglandin E2 (PGE2)**	LNCaP	Prostate cancer	Decreased proliferation and malignancy of LNCaP cells and caused the suppression of the expression of oncogenes such as c-myc and Bcl-2 in differentiated LNCaP cells.	[74]
**Papaverine**	PC-3	Prostate cancer	Showed cytotoxic effects by inducing early and late apoptosis along with inducing sub-G1 cell cycle arrest, and caused the downregulation of Blc-2, Bax, and NF-kB proteins and PI3K and phospho-Akt expression.	[75]
**Papaverine**	HT29, T47D, and HT1080	Colorectal cancer, breast cancer, and fibrosarcoma cells	Showed cytotoxic effects by selective DNA damage and induction of apoptosis.	[76]
**Papaverine**	MCF-7 and MDA-MB-231	Breast cancer	Showed cytotoxic effects by arresting cell cycle in G_0_/G_1_ phase and inducing apoptosis.	[77]
**Papaverine**	HepG-2	Hepatocarcinoma	Induced antiproliferative activity by inhibiting telomerase through downregulation of hTERT gene.	[78]
**Papaverine combined with temozolomide**	U87MG and T98G	Glioblastoma	Significantly inhibited the clonogenicity of the cell lines, delayed tumor growth, and increased the radiosensitivity of T98G cells.	[80,81]
**Papaverine–Au(III) complex**	MCF-7 and HepG-2	Breast cancer and hepatocellular carcinoma	Showed significant cytotoxic activity against the examined cell lines. Additionally, the Au complex showed anticancer activity against the breast cancer MCF-7 cells better than that of cisplatin.	[16]
**Papaverine**	HCT15 (colon), A549 (lung), HeLa (cervical), K562 (Bcr-Abl positive CML), and RAW 264.7	Colon, lung, cervical, and lymphoblast cancers	Induced ROS-mediated apoptosis and inhibited Bcr-Abl downstream signaling.	[84]
**Caroverine, derivative of papaverine**	LT97 and SW480	Colorectal cancer	Inhibition of expression of VEGF.	[82]
**6a,12a-diazadibenzo-[a,g]fluorenylium, derivative of papaverine**	MCF-7	Breast cancer	Inhibition of MCF-7 cell line by blocking G0/G1 phase of the cell cycle and telomerase activity.	[83]

## Data Availability

Not applicable.

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
