# Peer review of "Papaverine: A Miraculous Alkaloid from Opium and Its Multimedicinal Application"

_molecules, 2023, doi:10.3390/molecules28073149_

Round 1

Reviewer 1 Report

The review by B. Kim et al. deals with well-known alkaloid papaverine and its pharmacological applications. A very short description of the chemistry and biosynthesis studies of papaverine is also included.

In my opinion there are some criticisms to be moved to the authors.

First, several reviews on pharmacological potential of papaverine have been appeared in the literature in the last two years covering almost all pharmacological applications. The present manuscript should introduce some elements of novelty to the literature avoiding to be only a repetition of published papers. 

Second, the chemistry and biosynthesis sections could be good points of discussion making the review quite different from those published but, unfortunately, the content of both paragraphs is very poor (especially biosynthesis paragraph). These paragraphs should be completely re-written and significantly improved.

The entire manuscript should be improved with regards to the presentation.

Thus, I think that the paper is not worthy to be considered for publication in Molecules, at least in the present form.

Author Response

First, we are thanking Reviewer 1 for the valuable suggestions to improve the overall quality of the existing manuscript. We have revised the manuscript point-by-point as per the suggestion of the reviewer. Please find the Authors’ responses below:

Reviewer 1

  1. Several reviews on pharmacological potential of papaverine have been appeared in the literature in the last two years covering almost all pharmacological applications. The present manuscript should introduce some elements of novelty to the literature avoiding to be only a repetition of published papers.

Authors’ Reply: In the manuscript, we have presented an overall profile of Papaverine that can be considered a remarkable natural source of potential medicine having multilevel biological activities. Here are some differences between our manuscript and other published ones.

  1. In our manuscript, we have studied the underlying mechanism of the biological activities of Papaverine published in numerous reports and have demonstrated the molecular mechanism in different figures and tables to make it more palatable to the readers.
  2. We have accumulated two separate biosynthetic pathways of papaverine and presented them as steps of reactions including the enzymes necessary for each step. This point has already been acknowledged by you. (We have modified it further).
  • We have tried to link up the information we got from different articles and presented it in an unbiased manner. For example, in one article we got Papaverine produces side effects like priapism, in another article we got when papaverine is combined with phentolamine and PGE1, priapism can be subsided. We have correlated both the ideas we got from separate articles. (Line 191-195),
  1. The manuscript displays a wholesome picture of papaverine including its source, biosynthesis, and proofs of biological activity (in vitro, in vivo, and most importantly numerous clinical studies done on the phytochemical).

  1. The chemistry and biosynthesis sections could be good points of discussion making the review quite different from those published but, unfortunately, the content of both paragraphs is very poor (especially biosynthesis paragraph). These paragraphs should be completely re-written and significantly improved.

Authors’ Reply: We thank the reviewer for the scholarly suggestion. We have modified both chemistry and the biosynthesis paragraph. We have explained both biosynthetic pathways in the most detailed manner possible. Line 118-144.

  1. The entire manuscript should be improved with regards to the presentation.

Authors’ Reply: We have checked the whole manuscript and modified accordingly.

Reviewer 2 Report

In the article  “Papaverine: A miraculous alkaloid from opium and its multimedicinal application” the authors draw attention on benzylisoquinoline alkaloid which can be obtained from Papaverine somniferum L. (Opium poppy). Opium alkaloids were used in various conditions in Chinese and Indian herbal medicine. 

These mechanisms are very important for medical perspectives in therapeutically applications support.

There are several points that need to be addressed

The abstract maybe to be completed with some data about the molecular mechanisms described in the work.

Line 62: Please insert PDE (phosphodiesterase) inhibitors. 

Paragraph 2: I suggest improving the paragraph about natural resource of papaverine, adding supplementary information about diversity of Papaverus genus and maybe worldwide cultivation.

I recommend improve text editing: 

Line 311 and Table 2 Please insert caroverine correction.

Line 315 Insert the italic styles for scientific name S. cerevisiae.

Please use the italic style for in vivo and in vitro. 

The authors may could improve the article by including additions suggested. 

Author Response

First, we are thanking  Reviewer 2 for the valuable suggestions to improve the overall quality of the existing manuscript. We have revised the manuscript point-by-point as per the suggestion of the reviewer. Please find the Authors’ responses below:

Reviewer 2

1. The abstract maybe to be completed with some data about the molecular
mechanisms described in the work.
Authors’ Reply: We thank the reviewer for the thoughtful suggestion. The
basic molecular mechanism of Papaverine has been added to the abstract.

2. Line 62: Please insert PDE (phosphodiesterase) inhibitors.
Authors’ Reply: The information has been added.

3. Paragraph 2: I suggest improving the paragraph about natural resource of
papaverine, adding supplementary information about diversity of Papaverus genus
and maybe worldwide cultivation.
Authors’ Reply: We thank the reviewer for the thoughtful suggestion. The
section has been improved as per the suggestion.

4. Line 311 and Table 2 Please insert caroverine correction.
Authors’ Reply: The spellings have been corrected.

5. Line 315 Insert the italic styles for scientific name S. cerevisiae.
Authors’ Reply: The formatting has been done.

6. Please use the italic style for in vivo and in vitro.
Authors’ Reply: The whole manuscript has been rechecked and the
necessary corrections are made.

Reviewer 3 Report

Dear author(s):

Papaverine: A miraculous alkaloid from opium and its multimedicinal application

After an exhaustive revision, the manuscript is Accept in present form. In general, the study is closely connected to the journal's objectives. The study is very interesting. The English is good, and the authors have directed the review in the right way.

Author Response

We are thanking reviewer 3 for acknowledging our work and accepting it. 

With best regards,

Corresponding author.

Round 2

Reviewer 1 Report

Paragraphs 3 and 4 have been improved.

Pag. 3, lines 109-114

The authors should report correctly the spectroscopic methods cited in ref. 20 (i.e. MALDI Orbitrap imaging mass spectrometry means Matrix-assisted laser desorption/ionization coupled with Orbitrap imaging mass spectrometry).  “MALDI” and “Orbitrap imaging mass spectrometry” describe a single MS method.

DFT in this case means: density functional theory.

Please check ref. 20: “… to report here the theoretical and experimental elucidation of the papaverine biosynthetic pathway via (S)-reticuline, using advance quantum chemical DFT calculations, as well as experimental electronic absorption (EAs), diffuse reflectance (Ds) and fluorescence (Fs) spectroscopy, circular dichroic (CD) and MALDI Orbitrap imaging mass spectrometry (MS).”

Author Response

First, we are thanking Reviewer 1 for the valuable suggestions to improve the overall quality of the
existing manuscript. We have revised the manuscript point-by-point as per the suggestion of the
reviewer. Please find the Authors’ responses below:
Reviewer 1

1. Pag. 3, lines 109-114
The authors should report correctly the spectroscopic methods cited in ref.
20 (i.e. MALDI Orbitrap imaging mass spectrometry means Matrix-
assisted laser desorption/ionization coupled with Orbitrap imaging mass
spectrometry). “MALDI” and “Orbitrap imaging mass spectrometry”
describe a single MS method.
Authors’ Reply: We thank the reviewer for the scholarly advice. The
sentence has been corrected.

2. DFT in this case means: density functional theory.
Authors’ Reply: We thank the reviewer for the scholarly advice. The
sentence has been corrected.